# The VP3 Protein of Bluetongue Virus Associates with the MAVS Complex and Interferes with the RIG-I-Signaling Pathway

**DOI:** 10.3390/v13020230

**Published:** 2021-02-02

**Authors:** Marie Pourcelot, Rayane Amaral Moraes, Aurore Fablet, Emmanuel Bréard, Corinne Sailleau, Cyril Viarouge, Lydie Postic, Stéphan Zientara, Grégory Caignard, Damien Vitour

**Affiliations:** UMR 1161 Virologie, Laboratory for Animal Health, INRAE, Department of Animal Health, Ecole Nationale Vétérinaire d’Alfort, ANSES, Université Paris-Est, 94700 Maisons-Alfort, France; mariepourcelot@hotmail.com (M.P.); rayane.amaralmoraes@anses.fr (R.A.M.); aurore.fablet@vet-alfort.fr (A.F.); emmanuel.breard@anses.fr (E.B.); corinne.sailleau@anses.fr (C.S.); Cyril.VIAROUGE@anses.fr (C.V.); lydie.postic@anses.fr (L.P.); stephan.zientara@vet-alfort.fr (S.Z.); gregory.caignard@inrae.fr (G.C.)

**Keywords:** Bluetongue virus, VP3, MAVS, IKKε, type-I interferons, virus–host interactions

## Abstract

Bluetongue virus (BTV), an arbovirus transmitted by *Culicoides* biting midges, is a major concern of wild and domestic ruminants. While BTV induces type I interferon (alpha/beta interferon [IFN-α/β]) production in infected cells, several reports have described evasion strategies elaborated by this virus to dampen this intrinsic, innate response. In the present study, we suggest that BTV VP3 is a new viral antagonist of the IFN-β synthesis. Indeed, using split luciferase and coprecipitation assays, we report an interaction between VP3 and both the mitochondrial adapter protein MAVS and the IRF3-kinase IKKε. Overall, this study describes a putative role for the BTV structural protein VP3 in the control of the antiviral response.

## 1. Introduction

The Bluetongue virus (BTV) belongs to the *Orbivirus* genus within the *Reoviridae* family. Its genome consists of 10 double-stranded RNA segments that encode seven structural proteins (VP1-7) and five nonstructural proteins (NS1-4, NS3A). Additionally, a putative new NS protein (S10-ORF2) has been described more recently [1,2,3,4]. BTV is responsible for the Bluetongue (BT) disease that affects domestic and wild ruminants and causes significant economic losses in infected livestock. Initially present in tropical and subtropical regions, the virus has gradually spread to the Mediterranean basin since the end of the 20th century. It regularly causes epizootic diseases of variable intensity, the most important of which remains the introduction of a new strain of serotype 8 in Northern Europe in 2006 [5,6].

Innate immunity is the body’s first line of defense against viral infections. At the cellular level, this principally results in the initiation of the type I interferon (mainly alpha/beta interferon (IFN-α/β)) response. This signaling pathway is initiated following the recognition of viral molecular signatures, mainly nucleic acids, by specialized receptors (pattern recognition receptors, PRRs) [7,8]. Among the multitude of PRRs involved in the antiviral response, membrane-bound Toll-like receptors (TLRs) and the cytosolic RNA helicases RIG-I and MDA5 are the most studied. Once activated by nucleic acid recognition, they undergo a conformational change allowing the exposure of their amino-terminal caspase activation and recruitment domain (CARD), which will then form a homotypic interaction with the CARD domain present on the mitochondrial adaptor protein mitochondrial activating signaling protein (MAVS) [9,10,11,12]. It serves as a platform to orchestrate the activation of several signaling cascades to downstream factors, including tumor necrosis factor (TNF) receptor-associated factors (TRAFs) proteins and kinases (i.e., TBK1, IKKε, and the IKK complex), which in turn will phosphorylate and activate transcription factors (i.e., IFN regulatory factor (IRF3) and IRF7 or nuclear factor κB (NF-κB) [13,14,15]. These transcription factors are responsible for the production of IFN-α/β and other inflammatory cytokines [16]. The synthesis and release of IFN-α/β in the extracellular compartment will then initiate a second signaling cascade, the so-called Jak/STAT pathway, which leads to the synthesis of hundreds of genes with antiviral or immunomodulatory properties.

BTV is a strong inducer of IFN-I both in vivo and in vitro in multiple cell types derived from various tissues and species. We identified RIG-I and MDA5 helicases as sensors of BTV infection in nonhematopoietic cells and found that these molecules displayed antiviral activities against the virus [17]. Other PRRs are also able to sense BTV infection; indeed, an MyD88-dependent TLR-independent pathway is preferentially engaged in plasmacytoid dendritic cells (pDCs) [18]. As IFN-I is detrimental for viral replication, most viruses have put in place regulatory mechanisms. During BTV infection, both the induction and the response pathway of the IFN-signaling cascade are targeted by the virus [19,20]. The nonstructural BTV NS3 protein specifically interferes with IFN synthesis by targeting optineurin (OPTN), a resident protein of the Golgi apparatus that is essential for TBK1 activation and subsequent type I IFN-signaling [20,21]. At the response level, we showed that BTV impairs STAT phosphorylation and reduces the amount of JAK1 and TYK2 [19]. Importantly, Avia et al. recently demonstrated that NS3 interacts with STAT2 to trigger its degradation in an autophagy-dependent manner [22]. In addition, NS4 and the putative S10-OR2/NS5 proteins induce a global cellular shutoff, which may also contribute to evading the host intrinsic antiviral response [2,3]. In a previous study, significant inhibition of the IFN-β promoter activity by other BTV proteins, including the core protein VP3, was noticed [20]. Together with VP7, VP3 constitutes the inner capsid of the BTV particle. It plays an essential function in viral RNA replication and virion assembly. In BTV-infected cells, VP3 colocalizes with NS2 within virus inclusion bodies, where viral replication takes place [23]. In this paper, we investigated the molecular mechanisms that may explain the action of VP3 on the induction of IFN-α/β synthesis.

## 2. Materials and Methods

### 2.1. Cell Lines

HEK293T and HeLa cells were grown in Dulbecco’s modified Eagle’s medium (DMEM) supplemented with 10% heat-inactivated fetal calf serum (FCS), 1% pyruvate, and 1% penicillin and streptomycin (*P*-S). Cells were maintained at 37 °C in a humid atmosphere of 5% CO_2_ and air.

### 2.2. DNA Transfection and Plasmids

HEK293T and HeLa cells were transfected using Polyplus transfection (JetPRIME), according to the manufacturer’s instructions. ORFs-encoding sequences from BTV8 WT strain were cloned into dedicated destination vectors as described in [24]. GST-tag and 3xFlag-tag fusions were achieved by Gateway cloning using pDEST27 (Thermo Fisher Scientific) and pCI-neo-3xFlag vector, respectively. An expression vector pNRIG-I carrying genes for the constitutively active N-terminal CARDs of human RIG-I (NRIG-I) has been used to stimulate the luciferase reporter gene downstream of an IFN-β specific promoter sequence as previously described [25]. pcDNA3.1-Flag-TRAF3, pDEST27-MAVS, pcDNA3.0-MAVS-c-Myc and all MAVS mutant constructs were kindly obtained from Eliane F. Meurs (Institut Pasteur, Paris). Moreover, finally, pSPICA-N1 human TRAF3, MAVS, TBK1, IKKε and IRF3, as well as pSPICA-N1-RRS, were generously provided by Yves Jacob and Caroline Demeret (Institut Pasteur, Paris).

### 2.3. Reporter Luciferase Assay

24-well plates were seeded with 3 × 10^5^ HEK293T cells per well. The day after, cells were transfected with the empty pCI-neo-3xFlag expression vector or encoding viral proteins as specified. Cells were jointly co-transfected with 100 ng of firefly luciferase constructs under the control of the IFN-β or the ISG56 promoter and 10 ng of the pRL-CMV reference plasmid (Promega, Madison, WI, USA). Finally, 300 ng of pNRIG-I was transfected to activate the indicated promoters. The total amount of DNA was equally adjusted with pCI-neo-3xFlag. After 40 h, transfected cells were collected, and both firefly and *Renilla* luciferase activities were determined using the Bright-Glo and *Renilla*-Glo luciferase assay system (Promega), respectively. All graphs represent mean ratios between luciferase and *Renilla* of triplicate samples and include error bars of the standard deviation.

### 2.4. *Gaussia* Princeps Complementation Assay (GPCA)

White 96-well plates were seeded with 3.5 × 10^4^ HEK293T cells per well. After 24 h, cells were transfected using JetPRIME with 100 ng of pSPICA-N2-VP3 BTV8 and 100 ng of pSPICA-N1-IFN-β pathway agonists. At 24 h post-transfection, cells were lysed with 40 µL of *Renilla* lysis buffer (Promega) for 30 min. *Gaussia princeps* luciferase activity was assessed by injecting 50 µL of luciferase substrate reagent (Promega) on a FLUOstar Optima (BMG Labtech, Ortenberg, Germany). Results were expressed as a fold change normalized over the sum of controls, specified as normalized luminescence ratio (NLR) [26,27]. For a given protein pair A/B, NLR = (Gluc1-A + Gluc2-B)/[(Gluc1-A + Gluc2) + (Gluc1 + Gluc2-B)]. The NLR of the protein pair was considered as “validated” if above the upper limit of the positivity threshold defined for a random reference set (RSS)/Bait pairs. The RRS contains 14 proteins, supposedly nonbinders proteins, and was kindly provided by Caroline Demeret and Yves Jacob (Institut Pasteur) [26]. To determine the positivity threshold, we calculated the mean and the standard deviation of the 14 Gluc2-VP3 BTV8 + Gluc1-RRS values and set the positivity threshold for the interactions as mean + (2× standard deviation), corresponding to a 97% confidence interval (CI: 97%).

### 2.5. Co-Affinity Purification

6-well plates were seeded with 1.5 × 10^6^ HEK293T cells per well. After 24 h transfection of the indicated plasmids, cells were lysed in lysis buffer (20 mM MOPS-KOH pH 7.4, 170 mM of KCl, 0.5% Igepal, 2 mM β-Mercaptoethanol), supplemented with complete protease inhibitor cocktail (Roche) for 20 min at 4 °C. For co-immunoprecipitation experiments, after centrifugation at 10,000× *g* for 20 min, the lysates were precleared with protein G-conjugated Sepharose 4-B beads (Sigma-Aldrich, Lyon, France) for 1 h and then incubated with specific antibody and fresh protein G-sepharose beads for 2 h at 4 °C. For GST pulldown assay, protein extracts were incubated for 2 h at 4 °C with 35 μL of glutathione–Sepharose beads (Amersham Biosciences, Buckinghamshire, United Kingdom) to purify GST-tagged proteins. Finally, beads were washed with ice-cold lysis buffer 3 times for 5 min, and protein complexes were denatured and resolved by SDS–PAGE. Each pulldown experiment described here has been repeated at least three times.

### 2.6. Western Blot Analysis

Purified complexes and protein extracts were resolved by SDS-polyacrylamide gel electrophoresis (SDS–PAGE) on 4–12% NuPAGE Bis–Tris gels with MOPS running buffer and transferred to a nitrocellulose membrane (Thermo Fisher Scientific, Waltham, MA, USA). Immunoblot analysis was performed with specific antibodies to detect endogenous MAVS (AT107; Alexis Biochemicals, Lausanne, Switzerland), actin (AC40; Sigma-Aldrich, Lyon, France). 3xFlag, c-Myc and GST-tagged proteins were detected with a mouse monoclonal HRP-conjugated anti-Flag (M2; Sigma-Aldrich, Lyon, France), a mouse monoclonal anti-c-Myc (9E10; Roche, Basel, Switzerland) and a rabbit polyclonal anti-GST antibodies (Sigma-Aldrich, Lyon, France), respectively. The antigen–antibody complexes were visualized by chemiluminescence (Clarity^TM^ Western ECL, Bio-Rad, Marnes-la-Coquette, France).

### 2.7. Fluorescence Microscopy

24-well plates (Ibidi μ-plates, BioValley, Illkirch-Graffenstaden, France) were seeded with 10^5^ HeLa cells. 24 h post-transfection of pCherry-C1, pCherry-C1-VP7 BTV8 or pCherry-C1-VP3 BTV8, cells were fixed by incubation in 4% paraformaldehyde solution (Electron Microscopy Sciences, Hatfield, PA, USA) for 20 min, and then treated with PBS-glycine (0.1 M) and PBS-Triton (0.5%) for 5 min and 15 min, to quench and permeabilize the cells, respectively. After 3 washing steps in PBS, nonspecific binding sites were blocked by incubating cells in a solution of 1% PBS-BSA in PBS for 1 h. Cells were then incubated for 1 h at room temperature with the specific MAVS (AT107; Alexis Biochemicals, Lausanne, Switzerland) antibody. They were washed three times in PBS and were finally incubated for 1 h with a PBS-BSA 1% solution containing the dye Hoechst 33.258 and secondary antibody (anti-rabbit/A11035; Thermo Fisher Scientific, Waltham, MA, USA). Preparations were visualized using a 516 Axio observer Z1 fluorescence inverted microscope (Zeiss, Iena, Germany).

## 3. Results

In a previous study, we showed that BTV NS3 exhibits a strong antagonistic activity against type I IFN synthesis using a luciferase reporter assay [20]. In the same work, VP3 and VP4 were also able to dampen the IFN-β promoter activity, but to a lesser extent. Here, we focus on the inhibitory impact of VP3 on IFN-β promoter activation.

First, HEK293T cells were transfected with an IFN-β reporter plasmid (pIFN-β-luc), a control vector encoding the *Renilla* luciferase under the control of a CMV promoter, which was used as a control to normalize the transfection efficiency, a plasmid expressing a constitutively active form of RIG-I (pNRIG-I) and increasing amounts of a 3xFlag-VP3 encoding plasmid. As shown in Figure 1A, VP3 dampened IFN-β promoter activity triggered by NRIG-I in a dose-dependent manner. In parallel, we also found that VP3 inhibits the activation of the ISG56 promoter (Figure 1B), a natural IFN-stimulated gene, indicating that VP3 is able to impair activation of specific promoters along the entire IFN pathway induced upon activation of the RLR-signaling cascade.

Many viral antagonists of the IFN pathway are able to interact with components of the RIG-I-signaling cascade to impair IFN synthesis [28]. In order to decipher the molecular mechanism responsible for the inhibitory effect of VP3, we assessed its putative interaction with components of the RIG-I pathway through two distinct methodological approaches: split-luciferase assay and GST-pulldown. We first used a split luciferase reporter assay in which the N-terminal part of the *Gaussia princeps* luciferase (Gluc2) was fused to VP3 while IFN agonists were expressed in fusion with the C-terminal of *Gaussia princeps* luciferase (Gluc1). Luciferase activity was assessed 24 h post-transfection of the dedicated expression constructs in HEK293T cells. Results were expressed as normalized luminescence ratio (NLR) of the detected level of luciferase as described in the Materials and Methods section and in [26]. The NLR accounts for the background GPCA signal, which was measured for each binding partner as described above. In addition, the luminescence values obtained with the RRS and VP3 protein were used to calculate a confidence interval. Thus, we considered a protein pair as interacting if the NLR was above the NLR threshold and above the upper limit of the confidence interval calculated from the RRS. As shown in Figure 2A, the NLR threshold was 15.2. MAVS and IKKε constructs both gave values above the NLR threshold (17.4 and 17.7, respectively), while all other constructs remained negative. In order to confirm these results, GST-tagged VP3 was expressed in HEK293T cells together with Flag-tagged proteins of the RIG-I pathway and purified two days later using glutathione–Sepharose beads (Figure 2B). IKKε and, to a lesser extent, MAVS, which also displayed a residual binding affinity for GST, were specifically co-purified with the GST-VP3. In contrast, TBK1 was barely detectable, confirming the results obtained with the GPCA assay. Overall, these results demonstrate that BTV VP3 associates with MAVS and IKKε.

Since GST is residually bound to MAVS, we investigated the specificity of the interaction between VP3 and MAVS by co-immunoprecipitation and fluorescence experiments at both ectopic and endogenous levels. In a primary experiment, a c-Myc-MAVS construct was overexpressed together with Flag-tagged proteins BTV VP3, VP7 or NS2. VP7 and NS2 were used as controls, as they did not inhibit RIG-I-mediated IFN promoter activity and are therefore not supposed to interact with components of the IFN pathway [20]. Upon c-Myc coprecipitation assay, VP3 was the only viral protein that specifically interacts with MAVS (Figure 3A). Then, to determine whether endogenous MAVS could bind to VP3, a specific MAVS antibody was used to perform an immunoprecipitation experiment in cells overexpressing Flag-tagged viral proteins, as shown in Figure 3B. Again, only ectopic VP3 was able to associate with endogenous MAVS (Figure 3B), confirming the specific interaction between both proteins.

Then, we assess the effect of VP3 on MAVS distribution using fluorescent microscopy. In cells transfected with pCherry-VP3, endogenous MAVS was partly relocalized in VP3 expressing foci, while MAVS distribution remained unmodified in pCherry-VP7 transfected cells (Figure 3C). Whether VP3 and MAVS colocalize at the mitochondria remains further examinations.

Finally, we wanted to precise the domain of interaction of VP3 on MAVS. We thus used multiple MAVS deletion mutants as depicted in Figure 4A and proceeded to coprecipitation assays. A minimal domain containing the C-terminal end of MAVS (a.a. 470–540) retained the ability to bind VP3 (Figure 4B). MAVS associates with K63-linked polyubiquitinated TRAF3 via its carboxy-terminal 455-PEENEY-460 consensus site, which in turn recruits and activates downstream kinases TBK1/IKKε [29]. In addition, MAVS-TRAF6 interaction leads to the activation of NF-κB and IRF7 [12,30,31]. VP3 could alter the complex formation of either MAVS or IKKε with upstream or downstream partners, including TRAFs proteins, to shut down further IFN-signaling. The molecular mechanisms of VP3’s action towards MAVS and IKKε will be further investigated in a future study.

In conclusion, our data suggest that VP3 of BTV interferes with the induction of the innate immune response by partially inhibiting the RLR-dependent signaling pathway. A balance most likely exists between the ability of the host to trigger the production of IFN-I and respond to it versus the ability of viruses to block this antiviral signaling pathway. A slight change in this precarious equilibrium can favor either the host or the virus and thus have an impact on viral pathogenesis. These preliminary data showed that VP3 interacts with key components of the RIG-I-signaling pathway, MAVS and IKKe, possibly explaining its inhibitory role on RIG-I-mediated IFN-β induction. A better understanding of the mode of action of VP3 to counteract the host response will give essential insights into the pathogenesis associated with BTV infection and will help to generate better prophylactic tools to control the virus.

## Figures and Tables

**Figure 1 viruses-13-00230-f001:**
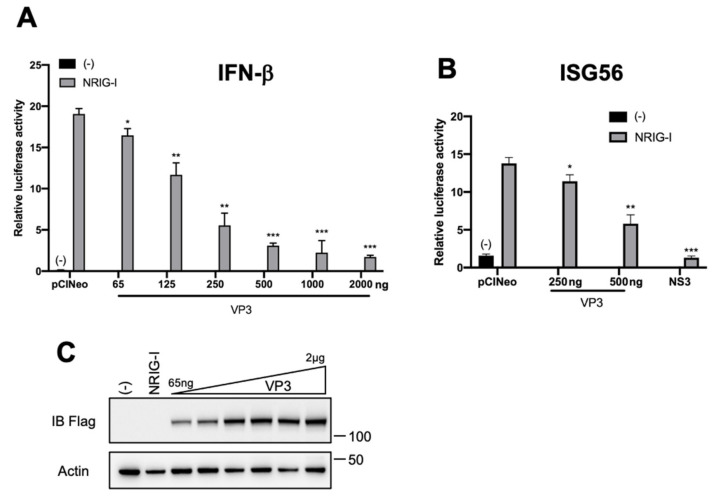
Bluetongue virus (BTV) VP3 inhibits the RIG-I-signaling pathway. HEK293T cells were transfected with 250 ng of the control plasmid (pCINeo-3xFlag) or increasing amounts of a plasmid carrying genes for 3xFlag-VP3 as indicated or 250 ng 3xFlag-NS3 of BTV8 together with 100 ng of the IFN-β-pGL3 (**A**) or the ISG56-pGL3 (**B**) plasmid that contains the firefly luciferase reporter gene downstream IFN-β and ISG56 promoter sequences, respectively, 300 ng of empty (-) or pNRIG-I plasmids, and 10 ng of the pRL-CMV reference plasmid. After 40 h, the relative luciferase activity was determined. Mean ratios between luciferase and *Renilla* activities of triplicate samples (± SD) are represented. *, *p* < 0.05; **, *p* < 0.005; ***, *p* < 0.0005 compared to pCINeo (**A**,**B**) (unpaired *t*-tests with Welch’s correction). (**C**) The expression of 3xFlag-VP3 and actin in (**A**) was determined in cell lysates by SDS–PAGE and immunoblotting with 3xFlag and actin antibodies.

**Figure 2 viruses-13-00230-f002:**
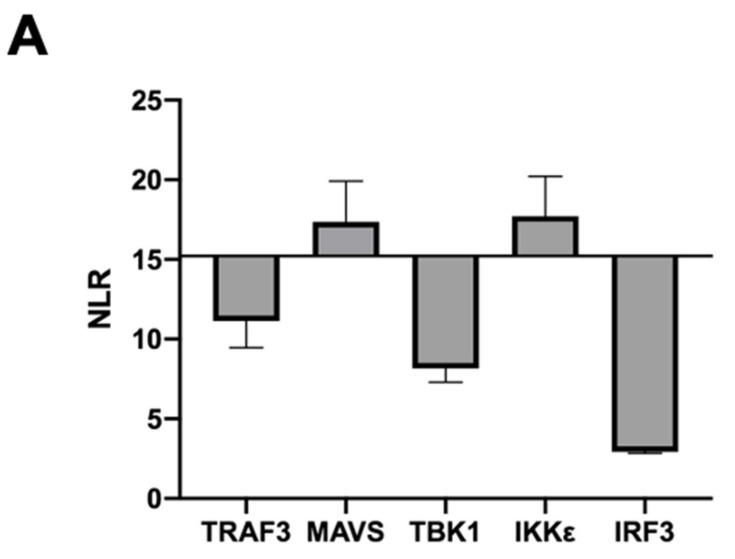
Interaction of VP3 with components of the mitochondrial activating signaling protein (MAVS) complex. (**A**) HEK293T cells were transfected with pSPICA-N2-BTV VP3 and pSPICA-N1-TRAF3, MAVS, TBK1, IKKε or IRF3. 24 h after transfection, the *Gaussia princeps* luciferase activity was measured. Representative normalized luminescence ratio (NLR) values for each protein pair are shown. Positive threshold was above 15. (**B**) HEK293T were co-transfected with plasmids encoding Gluathione S-Transferase (GST) alone or fused to BTV-VP3, and pCI-neo-3xFlag encoding 3xFlag tag fused to TRAF3, MAVS, TBK1, IKKε or IRF3. 24 h after transfection, co-purifications of interferon (IFN) pathway components were assayed by pulldown (PD) using glutathione–Sepharose beads. The presence of overexpressed proteins in cell lysates and precipitated complexes was determined by SDS–PAGE and immunoblotting with 3xFlag and GST antibodies.

**Figure 3 viruses-13-00230-f003:**
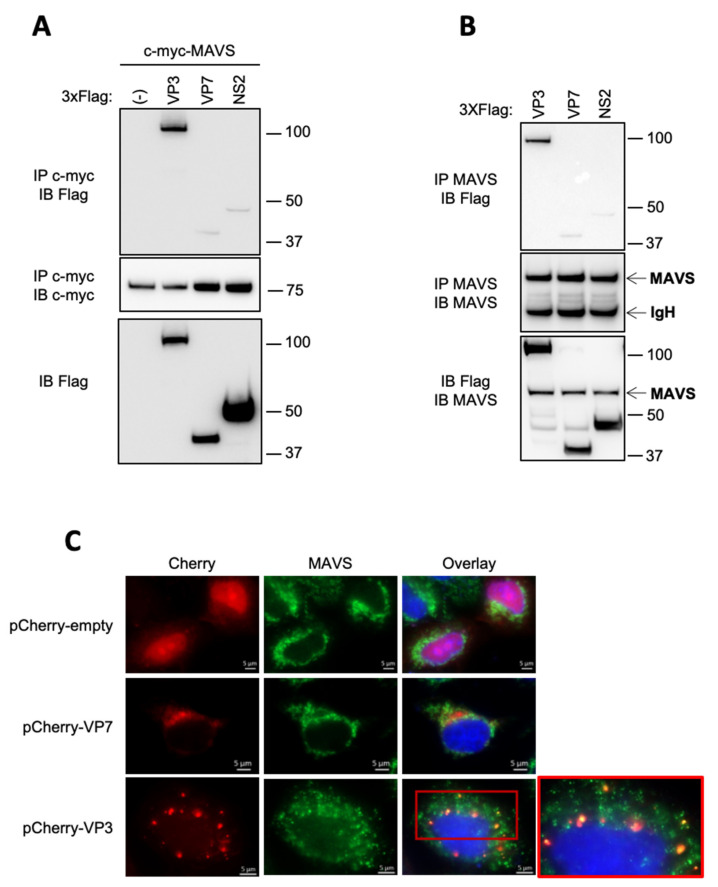
VP3 specifically interacts with MAVS. (**A**) HEK293T were co-transfected with pCI-neo-3xFlag encoding 3xFlag tag fused to BTV-VP3, BTV-VP7, BTV-NS2 or the corresponding empty vector (-) in the presence of pMyc-MAVS. 24 h after transfection, cell lysates were immunoprecipitated (IP) with anti c-Myc antibody. The immunoprecipitate and the cell lysate were resolved on SDS–PAGE and immunoblotted with anti 3xFlag and anti c-Myc antibodies. (**B**) HEK293T cells were transfected with plasmids encoding 3xFlag tag BTV-VP3, BTV-VP7 or BTV-NS2. 24 h after transfection, the cell lysates were immunoprecipitated (IP) with a specific antibody against endogenous MAVS. The immunoprecipitate and the cell lysate were resolved on SDS–PAGE and immunoblotted with anti 3xFlag and MAVS antibodies. (**C**) HeLa cells were transfected with pCherry-C1 plasmid encoding VP7 BTV8, VP3 BTV8 or the corresponding empty vector (-). 24 h after, cells were fixed and labeled with the dye Hoechst 33258 to stain nuclei and with specific antibodies for MAVS. Intracellular localization of Hoechst-stained nuclei (blue), BTV-VP3 and VP7 (red) and endogenous MAVS (green) were visualized by fluorescence microscopy (×63 magnification). Scale bars represent 5 µM.

**Figure 4 viruses-13-00230-f004:**
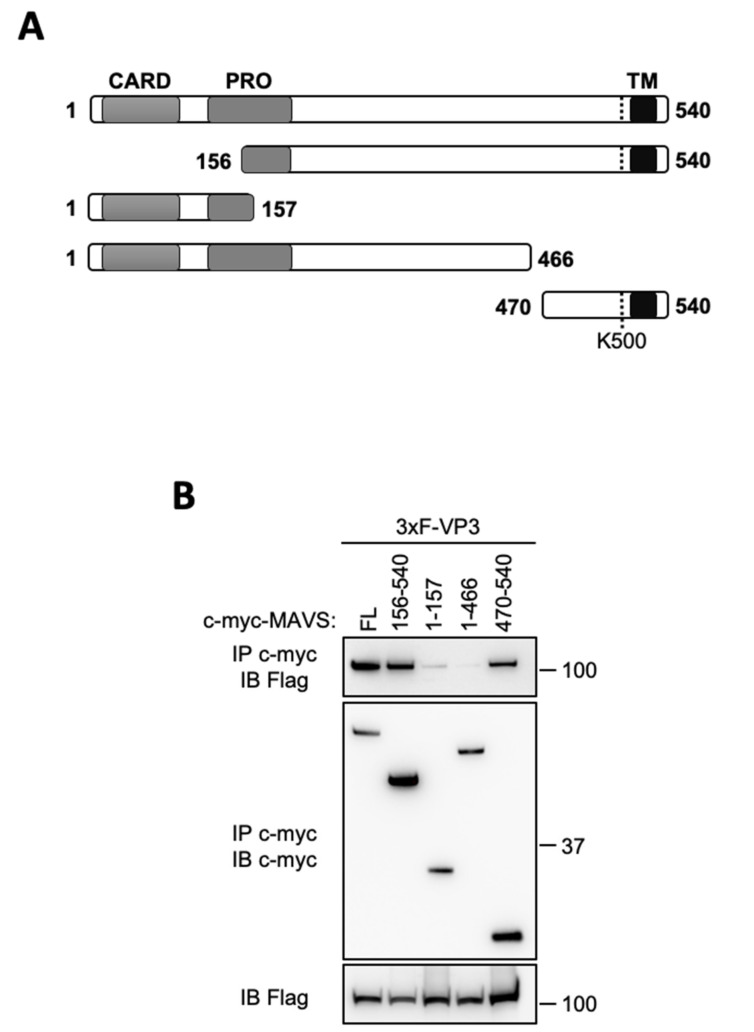
VP3 interacts with the C-terminal domain of MAVS. (**A**) Schematic representation of MAVS protein. The CARD domain (a.a. 1–77), the proline-rich region (PRO) (a.a. 103–173), the transmembrane domain (TM) (a.a. 514–535) and K500 residue are depicted. (**B**) HEK293T were transfected with a vector encoding 3xFlag-tagged BTV-VP3 in the presence of different c-Myc-MAVS constructs as indicated. 24 h after transfection, the cell lysates were immunoprecipitated (IP) with anti c-Myc antibody. The immunoprecipitate and the cell lysate were resolved on SDS–PAGE and immunoblotted with anti 3xFlag and anti c-Myc antibodies.

## Data Availability

All data are given in the Results section.

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
