# Peer review of "The VP3 Protein of Bluetongue Virus Associates with the MAVS Complex and Interferes with the RIG-I-Signaling Pathway"

_viruses, 2021, doi:10.3390/v13020230_

Round 1
Reviewer 1 Report
In this work, Pourcelot and colleagues try to address the molecular mechanisms implicated in the already described capacity of the VP3 BTV protein to inhibit type I interferon production. While the interaction between VP3 and MAVS is conclusively demonstrated by different approaches, and indeed, it constitutes the main (if not only) new message, the molecular mechanisms responsible for this inhibition remain obscure. The authors propose a competition with IKKe for the binding to MAVS, but a MAVS mutant that does not bind IKKe retains the capacity to bind VP3. As the inhibitory effect of VP3 is already known, some deeper mechanistic insights should be provided to enhance the novelty generated by this work.
MAJOR CONCERNS
1.- The inhibition generated by VP3 on the interaction between MAVS and IKKe, based on Figure 3B is not convincing to this reviewer. It looks like the amount of total MAVS (third plot from the top) is reduced when VP3 is transduced. If so, the reduction shown in the MAVS/IKKe could be due to a lower amount of the former. In other words, I would like to see the ratio between total MAVS (in a non-overexposed plot) and the recovered IKKe.
2.- Alternative ways of demonstrating this inhibition should be provided such as downstream kinase activation (for instance).
3.- All the protein-protein interaction experiments are based on overexpression experiments in HEK cells. Experiment using primary cells (macrophages) bearing the entire molecular machinery would be convenient to reinforce the new findings.
4.- In is indicated as “preliminary results”, the possibility of VP3 ubiquitination. As the molecular mechanism is the biggest gap in this work, the authors should dig into this putative mechanism.
MINOR CONCERNS
5.- MAVS intracellular distribution is affected by the presence of VP3 (Figure 4C). Could the authors discuss about it, trying to link this effect to the IFNI-inhibiting mechanism triggered by VP3?
6.- How many times were the pulldown experiments repeated? Please, indicate to know how representative are the images of the studied process.
7.- Statistical analysis is missing in figure 2B.
Author Response
Reviewers' comments:
Reviewer #1
Comments and Suggestions for Authors
In this work, Pourcelot and colleagues try to address the molecular mechanisms implicated in the already described capacity of the VP3 BTV protein to inhibit type I interferon production. While the interaction between VP3 and MAVS is conclusively demonstrated by different approaches, and indeed, it constitutes the main (if not only) new message, the molecular mechanisms responsible for this inhibition remain obscure. The authors propose a competition with IKKe for the binding to MAVS, but a MAVS mutant that does not bind IKKe retains the capacity to bind VP3. As the inhibitory effect of VP3 is already known, some deeper mechanistic insights should be provided to enhance the novelty generated by this work.
We agree with Reviewer#1 that the interaction between VP3 and MAVS is of great importance but we also provide evidence that VP3 could bind IKKe using split luciferase (Fig. 2A) and pulldown assays (Fig. 2B). Whether these interactions occur directly or indirectly within a complex remain to be further investigated in a future study. We also agree with Reviewer#1 that the inhibitory effect of VP3 on the interferon pathway has been suggested in a previous work (Chauveau et al., J Virol 2013) but this is the first original article that provides further molecular mechanisms regarding this effect.
MAJOR CONCERNS
1.- The inhibition generated by VP3 on the interaction between MAVS and IKKe, based on Figure 3B is not convincing to this reviewer. It looks like the amount of total MAVS (third plot from the top) is reduced when VP3 is transduced. If so, the reduction shown in the MAVS/IKKe could be due to a lower amount of the former. In other words, I would like to see the ratio between total MAVS (in a non-overexposed plot) and the recovered IKKe.
We agree with Reviewer#1 that the amounts of MAVS appear sometimes reduced in the presence of VP3. However, the total amount of MAVS expressed ectopically or at the endogenous level is not affected by VP3 overexpression (data not shown). Therefore, it seems that this finding is mostly related to the pulldown assay. We agree that this result may influence the interpretation of the effect of VP3 on the MAVS-IKKe complex. Indeed, VP3 could disrupt the formation of this complex and consequently induce the degradation of one and/or another of these actors who no longer find protection within this protein formation. Alternatively, we can also imagine that VP3 induces a change in conformation or a post-translational modification altering the stability of MAVS. Finally, VP3 could also recruit other factors that would play a role in protein stability. As the data presented in Fig 3 may appear not enough convincing and require further investigations, this figure has been removed from the revised version of the manuscript.
2.- Alternative ways of demonstrating this inhibition should be provided such as downstream kinase activation (for instance).
We plan to examine this point in more detail in a future study, including reporter assay, IRF3 activation at the endogenous level as well as the effect of VP3 in co-precipitation experiments with IKKe partners.
3.- All the protein-protein interaction experiments are based on overexpression experiments in HEK cells. Experiment using primary cells (macrophages) bearing the entire molecular machinery would be convenient to reinforce the new findings.
We agree with Reviewer#1 that it would have been optimal to test these hypotheses in a cellular model closer to the physiological conditions of infection, like macrophages, and also at the endogenous level but unfortunately, we did not benefit from specific antibodies directed against VP3 suitable for this type of experiments.
4.- In is indicated as “preliminary results”, the possibility of VP3 ubiquitination. As the molecular mechanism is the biggest gap in this work, the authors should dig into this putative mechanism.
As we do not have sufficiently consolidated data on this point at this stage, it has been withdrawn in the revised manuscript.
MINOR CONCERNS
5.- MAVS intracellular distribution is affected by the presence of VP3 (Figure 4C). Could the authors discuss about it, trying to link this effect to the IFNI-inhibiting mechanism triggered by VP3?
We agree with Reviewer#1 that MAVS is partly relocalized in VP3 expressing dots in cells overexpressing pCherry-VP3. A sentence has been added in the text (lanes 198-200 of the new version). Of note, HHV-8 vIRF-1 interacts with MAVS and inhibits MAVS aggregation, which is essential for its antiviral activity (Hwang, 2016). We can imagine that VP3 sequesters MAVS to impair its activation and/or association with cellular partners.
6.- How many times were the pulldown experiments repeated? Please, indicate to know how representative are the images of the studied process.
Each pulldown experiment described here has been repeated at least three times and the blots displayed in this article are representative of the most frequently obtained results.
7.- Statistical analysis is missing in figure 2B.
Despite lack of statistical analysis for this figure, the selected western blots set is representative of the most frequent findings.
Reviewer 2 Report
The authors describe a new putative role of the structural protein VP3 of BTV in the modulation of IFN. They start with an inhibition of RIG1 pathway, then finding an association between VP3 and MAVS and IKKe; then a specific interaction between VP3 and MAVS, shown to be located at the C-terminal of MAVS. The finding of this new putative role of VP3 deserves publication. The autors can find my comments here:
Abstract:
- the induction of IFN by BTV infection is mentioned here in the abstract but not in the introduction. This should be included in the introduction as well.
- Here, VP3 is suggested as an antangosit of IFN “beta” synthesis. There are no data evidencing a specific effect on IFN”beta” but suggesting an overall effect on “IFN” signalling.
Introduction:
- explain the abbreviation TRAFs : Tumor necrosis factor receptor-associated factors (TRAFs)
- move some parts of the discussion into introduction (what is VP3?, role of NS3, STAT phosphorylation, etc.) (see comments on discussion
Results:
Figure 1 :
- meaning of “NT” ; explain why no NT is displayed in fig1 A. ;
- units of X-axes (fig1A and B)
-in the legend : the text following (A) (line 150) is not clear : maybe renumbering the figure with A, B and C (referring as to SDS-PAGE) try to keep coherence between “IB flag” (label in the figure) and 3xFlag (in the legend).
Discussion:
- lines 230-241: this is not discussion but should be part of the introduction section. The discussion should start with the data obtained in the experiments.
- Lines 242-243: again mix between introduction and discussion: the role of VP3 should be explained in the introduction and not in the discussion
Lines 248-251: again mix between introduction and discussion: thus explanation of the role of MAVS should also be moved into introduction
- lines 270-271: the perspectives of a therapeutic use if very far. It is highly speculative.
-It should be interesting to discuss the relative role of these mechanisms antagonist of the IFN pathways and of the stimulation of IFN by BTV infection: there must be a balance between the two mechanisms.
Author Response
Reviewer 2
Open Review
(x) I would not like to sign my review report
( ) I would like to sign my review report
English language and style
( ) Extensive editing of English language and style required
( ) Moderate English changes required
( ) English language and style are fine/minor spell check required
(x) I don't feel qualified to judge about the English language and style
|
Yes |
Can be improved |
Must be improved |
Not applicable |
|
|
Does the introduction provide sufficient background and include all relevant references? |
( ) |
( ) |
(x) |
( ) |
|
Is the research design appropriate? |
(x) |
( ) |
( ) |
( ) |
|
Are the methods adequately described? |
(x) |
( ) |
( ) |
( ) |
|
Are the results clearly presented? |
( ) |
(x) |
( ) |
( ) |
|
Are the conclusions supported by the results? |
( ) |
(x) |
( ) |
( ) |
Comments and Suggestions for Authors
The authors describe a new putative role of the structural protein VP3 of BTV in the modulation of IFN. They start with an inhibition of RIG1 pathway, then finding an association between VP3 and MAVS and IKKe; then a specific interaction between VP3 and MAVS, shown to be located at the C-terminal of MAVS. The finding of this new putative role of VP3 deserves publication.
We are grateful to Reviewer#2 for this positive comment.
The authors can find my comments here:
Abstract:
- the induction of IFN by BTV infection is mentioned here in the abstract but not in the introduction. This should be included in the introduction as well.
As suggested by Reviewer#2, we now further describe the IFN induction by BTV in the Introduction section of the revised manuscript which has been adapted to a Short communication format.
- Here, VP3 is suggested as an antangosit of IFN “beta” synthesis. There are no data evidencing a specific effect on IFN”beta” but suggesting an overall effect on “IFN” signalling.
We agree with Reviewer#2, that VP3 might alter not only IFN beta synthesis and could have a broader impact on IFN signaling. However, as we only used an IFN-beta -luciferase reporter assay to evaluate the putative inhibitory role of VP3 on IFN-I production, conclusions were drawn from results obtained with this promoter only.
Introduction:
- explain the abbreviation TRAFs : Tumor necrosis factor receptor-associated factors (TRAFs)
The text has been modified accordingly.
- move some parts of the discussion into introduction (what is VP3?, role of NS3, STAT phosphorylation, etc.) (see comments on discussion
The introduction section has been amended as requested and VP3 features have been further described (see lanes 64-66).
Results:
Figure 1 :
- meaning of “NT” ; explain why no NT is displayed in fig1 A. ;
“NT” is inappropriate as it corresponds to a negative control (empty plasmid) that has been used to compensate the amount of DNA used for the RIG-N agonist construct. Thus, “(-)” has been added on Fig 1A and B.
- units of X-axes (fig1A and B)
The unit (ng) has been added on the X-axes
-in the legend : the text following (A) (line 150) is not clear : maybe renumbering the figure with A, B and C (referring as to SDS-PAGE) try to keep coherence between “IB flag” (label in the figure) and 3xFlag (in the legend).
According to the suggestion of Reviewer#2, the legend of Figure 1 has been improved to be clear enough. The immunoblot is now numbered 1C.
Discussion:
The manuscript has been formatted as Short communication article, and therefore the discussion section has been removed and partly fused within the Results section. But as suggested by Reviewer#2, many points described below have been addressed and appeared now in the Introduction section.
- lines 230-241: this is not discussion but should be part of the introduction section. The discussion should start with the data obtained in the experiments.
We agree with Reviewer#2 and description of the type I IFN signaling cascade and the regulation of this pathway by BTV are depicted the Introduction section (lanes 31-67).
- Lines 242-243: again mix between introduction and discussion: the role of VP3 should be explained in the introduction and not in the discussion
Again, we agree with Reviewer#2 and few words on VP3 functions have been added in the Introduction section (lanes 63-66).
Lines 248-251: again mix between introduction and discussion: thus explanation of the role of MAVS should also be moved into introduction
According to the suggestion of Reviewer#2, the role of MAVS in the IFN signaling cascade has been mentioned in the Introduction section (lanes 36-44).
- lines 270-271: the perspectives of a therapeutic use if very far. It is highly speculative.
We agree with Reviewer#2 that the use of molecules targeting the MAVS-VP3 interaction for therapeutic purposes is speculative at this stage and the sentence has thus been removed.
-It should be interesting to discuss the relative role of these mechanisms antagonist of the IFN pathways and of the stimulation of IFN by BTV infection: there must be a balance between the two mechanisms.
As the format has been changed to Short communication, a sentence has been added in the Conclusion section (lanes 226-227): “A slight change in this precarious equilibrium can favor either the host or the virus and thus have an impact on viral pathogenesis.”
Reviewer 3 Report
In this study (viruses-950630), the authors tried to demonstrate that Bluetongue virus (BTV) subcore protein, VP3, could interact with mitochondrial antiviral-signaling protein (MAVS) to negatively regulate RIG-I signaling pathway. The data showed that BTV VP3 inhibited RIG-I dependent IFN-β/ISG56 promoter activities in a dose-dependent manner. Then, using pull-down assay and protein-complementation assay (PCA), MAVS and IκB kinase-ε (IKKε) were determined as targets for VP3 association. The authors focused on MAVS and the pull-down assay with a series of MAVS deletions showed that VP3-binding domain could be C-terminal region (470-540 aa) of MAVS.
It would be absolutely great if the inhibition mechanisms of RIG-I signaling pathway by BTV VP3 was clarified. The reviewer agrees that IFN responses via RIG-I signaling pathway could be blocked by VP3 and that tagged VP3 bound to either tagged MAVS or tagged IKKε. However, unfortunately, none of data shown in this manuscript provides enough evidence to prove the negative regulation of INF responses by "VP3-MAVS interaction". Their fundamental misunderstandings, regarding TRAF-mediated "positive & negative" regulation of IFN signaling via MAVS, misled their conclusions. The authors concluded that the VP3-MAVS association impairs the recruitment of IKKε to MAVS and consequently block the MAVS signaling pathway. However, the recruitment of IKKε to mitochondria via K63-linked polyubiquitination at K500 of MAVS (lanes 233-235, 276-278) is likely to initiate the signal to "negatively" regulate NF-κB (ref#20) and "shut down" the IFN response (Paz S et al 2011 Cell Res). Thus, blocking IKKε-MAVS (by VP3) should demolish "IKKε INF-shutdown pathway" to further induce IFN & ISGs. To activate IFN & ISG induction, MAVS recruits K63-linked polyubiquitinated TRAF3 to the C-terminal 455-PEENEY-460 consensus site to recruit and activate downstream kinases TBK1/IKKε (Paz S et al 2011 Cell Res). MAVS-TRAF6 interaction also leads to recruit and activate downstream kinases TBK1/IKKε (Konno H et al 2009 PLoS One, etc). VP3-MAVS interaction might impair the recruitment of TRAF3/6 to MAVS to shut down further activation of RIG-I pathway. Or VP3- IKKε interaction may be critical to shutdown IFN signaling. In any case, further experiments should be required.
The manuscript is mostly understandable however, there are typos, grammatical errors, misusages of terms and full of unprofessional English. Before submission, the manuscript should get some scientific English proofreading.
Unfortunately, this manuscript is not sufficient for Viruses level.
- The introduction is not informative. The introduction should supply enough background information and the rationale for the present study to allow the reader to understand and evaluate the results of the present study “without referring to previous publications on the topic”. It would be better to describe more about RIG-I pathway, especially MAVS and IKKε as this study focused on those elements.
- The text should be covering essential Materials & Methods. Some important information, such as immunoprecipitation, is completely missing. In addition, pCl-neo-3xFlag constructs for BTV-8 proteins have already published their previous paper, which was not referred (Kundlacz C et al 2019 J Virol). Thus, sentences between lanes 54 and 61, which are very similar to those in the previous publication, should be removed (except for construction of GST-tagged VP3 vector). Instead of those copy-pasted sentences, newly constructed vectors, such as pCl-neo-3xFlag-TRAF3, pDEST27-MAVS..., should be described. Further, for PCA using split Gaussia princeps luciferase (Gluc1 and Gluc2) fused with each of RIG-I pathway elements and BTV VP3, respectively, cutoff value of Normalized Luminescence Ratio (NLR) seemed to be wrongly set. According to ref#15, the random reference set (RRS) means "an a priori 'negative' or non-interacting set of 100 random human protein pairs" and the NLR cutoff for "non-interacting protein pairs" was set at 3.5. According to the manuscript, the authors did PCA using Gluc2-fused VP3 and Gluc1-fused RRS to determine cutoff value, which is likely to be around 15 (Fig. 2A). It is very hard to know what Gluc1-fused RRS is, however, it could be randomly selected human protein X. Please explain why the authors did not use the evaluated cutoff value (3.5) and how the authors knew that VP3 did not interact with protein X?
- 1: According to Materials & Methods, empty vector should be pCl-neo-3xFlag, not pCl-neo. Thus "a plasmid carrying gene for 3xFlag-VP3...." should be " a plasmid carrying gene for VP3...." Please amend legend.
- 2A: The authors tried to emphasize cutoff value, however, it is wired. Please amend it. in addition, actual cutoff value should be mentioned in the legend.
- Fig 3A and related text "BTV VP3 impairs the association between MAVS and IKKε": Disagree. The binding ability of IKKε to MAVS should be evaluated properly. For example, the relative density of flag- IKKε should be normalized by the relative density of myc-MAVS. Similarly, the relative density of flag- VP3 should be normalized by the relative density of myc-MAVS. In addition, each experiment should be at least triplicated. After statistical analysis, changes of IKKε binding ability by VP3 would be discussed. Sigle experiment is not acceptable. In addition, according to previous studies on BTV VP3 (Kar A et al 2004 Virology....), VP3 is likely to form decamers and easily aggregate in certain condition. It would be better to confirm that VP3 was not aggregated and precipitated nonspecifically during pull-down assay.
- 4 and related text: Why the authors chose VP4 and VP7? As NS3 is likely to be IFN antagonist, NS3 should be included.
- 5 and related text: The MAVS deletion mutant 1-446 did not exist in ref#13.
- Lanes 235-237: The authors logically failed to describe why they tried to confirm if VP3 could bind to K500 of MASV. MASV, "not IKKε", was K63-linked polyubiquitinated at K500 and IKKε interacted with the polyubiquitin on K500 (ref#20). Please explain why the authors thought that the ubiquitination of VP3 could be involved in MAVS interaction, logically.
- Lane 204: Ref#19 does not show "the recruitment domain of IKKε". There may be more misusages of references. Please check carefully.
- Discussion contains backgrounds, which should be written in Introduction, extensive repetition of the Results section, extensive future works, incorrect interpretation of results and full of speculations. Please provide much more interpretation of the results in relation to previously published work.
Author Response
Reviewer #2
Comments and Suggestions for Authors
In this study (viruses-950630), the authors tried to demonstrate that Bluetongue virus (BTV) subcore protein, VP3, could interact with mitochondrial antiviral-signaling protein (MAVS) to negatively regulate RIG-I signaling pathway. The data showed that BTV VP3 inhibited RIG-I dependent IFN-β/ISG56 promoter activities in a dose-dependent manner. Then, using pull-down assay and protein-complementation assay (PCA), MAVS and IκB kinase-ε (IKKε) were determined as targets for VP3 association. The authors focused on MAVS and the pull-down assay with a series of MAVS deletions showed that VP3-binding domain could be C-terminal region (470-540 aa) of MAVS.
It would be absolutely great if the inhibition mechanisms of RIG-I signaling pathway by BTV VP3 was clarified. The reviewer agrees that IFN responses via RIG-I signaling pathway could be blocked by VP3 and that tagged VP3 bound to either tagged MAVS or tagged IKKε. However, unfortunately, none of data shown in this manuscript provides enough evidence to prove the negative regulation of INF responses by "VP3-MAVS interaction". Their fundamental misunderstandings, regarding TRAF-mediated "positive & negative" regulation of IFN signaling via MAVS, misled their conclusions. The authors concluded that the VP3-MAVS association impairs the recruitment of IKKε to MAVS and consequently block the MAVS signaling pathway. However, the recruitment of IKKε to mitochondria via K63-linked polyubiquitination at K500 of MAVS (lanes 233-235, 276-278) is likely to initiate the signal to "negatively" regulate NF-κB (ref#20) and "shut down" the IFN response (Paz S et al 2011 Cell Res).
Thus, blocking IKKε-MAVS (by VP3) should demolish "IKKε INF-shutdown pathway" to further induce IFN & ISGs. To activate IFN & ISG induction, MAVS recruits K63-linked polyubiquitinated TRAF3 to the C-terminal 455-PEENEY-460 consensus site to recruit and activate downstream kinases TBK1/IKKε (Paz S et al 2011 Cell Res). MAVS-TRAF6 interaction also leads to recruit and activate downstream kinases TBK1/IKKε (Konno H et al 2009 PLoS One, etc). VP3-MAVS interaction might impair the recruitment of TRAF3/6 to MAVS to shut down further activation of RIG-I pathway. Or VP3- IKKε interaction may be critical to shutdown IFN signaling. In any case, further experiments should be required.
We agree with Reviewer#2 and we apologize for this misinterpretation of the possible role of VP3 on the MAVS-IKKeinteraction and the consequences on the modulation of the IFN synthesis. Moreover, the data presented in Fig 3 may appear not enough convincing as amounts of precipitated MAVS may lowered in some samples and thus skews interpretation. This figure has been removed from the revised version of the manuscript. In a future study, we plan to test the effect of VP3 on other complexes associating MAVS and IKKe partners, including TRAF proteins. A paragraph has been added in the discussion section to mention this perspective (lanes 256-263 of the new version).
The manuscript is mostly understandable however, there are typos, grammatical errors, misusages of terms and full of unprofessional English. Before submission, the manuscript should get some scientific English proofreading.
Unfortunately, this manuscript is not sufficient for Viruses level.
- The introduction is not informative. The introduction should supply enough background information and the rationale for the present study to allow the reader to understand and evaluate the results of the present study “without referring to previous publications on the topic”. It would be better to describe more about RIG-I pathway, especially MAVS and IKKε as this study focused on those elements.
The introduction has been modified accordingly, with more details on the components of the RIG-I pathway (see lanes 33-44 of the new version).
- The text should be covering essential Materials & Methods. Some important information, such as immunoprecipitation, is completely missing.
The Materials & Methods section has been further checked in the revised version.
- In addition, pCl-neo-3xFlag constructs for BTV-8 proteins have already published their previous paper, which was not referred (Kundlacz C et al 2019 J Virol). Thus, sentences between lanes 54 and 61, which are very similar to those in the previous publication, should be removed (except for construction of GST-tagged VP3 vector). Instead of those copy-pasted sentences, newly constructed vectors, such as pCl-neo-3xFlag-TRAF3, pDEST27-MAVS..., should be described.
The text has been modified accordingly (lanes 63-68).
- Further, for PCA using split Gaussia princeps luciferase (Gluc1 and Gluc2) fused with each of RIG-I pathway elements and BTV VP3, respectively, cutoff value of Normalized Luminescence Ratio (NLR) seemed to be wrongly set. According to ref#15, the random reference set (RRS) means "an a priori 'negative' or non-interacting set of 100 random human protein pairs" and the NLR cutoff for "non-interacting protein pairs" was set at 3.5. According to the manuscript, the authors did PCA using Gluc2-fused VP3 and Gluc1-fused RRS to determine cutoff value, which is likely to be around 15 (Fig. 2A). It is very hard to know what Gluc1-fused RRS is, however, it could be randomly selected human protein X. Please explain why the authors did not use the evaluated cutoff value (3.5) and how the authors knew that VP3 did not interact with protein X?
The luminescence values obtained with the RRS and VP3 protein were used to define a confidence interval. We considered a protein pair as interacting if the NLR was above the previously defined threshold of 3.5 (Cassonnet et al, Nat Methods, 2011) upper limit of the confidence interval calculated from the RRS (15 in the present study). Materials and methods and figure legend have been modified accordingly.
- 1: According to Materials & Methods, empty vector should be pCl-neo-3xFlag, not pCl-neo. Thus "a plasmid carrying gene for 3xFlag-VP3...." should be " a plasmid carrying gene for VP3...." Please amend legend.
The legend has been modified (lane 144 of the new version).
- 2A: The authors tried to emphasize cutoff value, however, it is wired. Please amend it. in addition, actual cutoff value should be mentioned in the legend.
The legend has been modified accordingly.
- Fig 3A and related text "BTV VP3 impairs the association between MAVS and IKKε": Disagree. The binding ability of IKKε to MAVS should be evaluated properly. For example, the relative density of flag- IKKε should be normalized by the relative density of myc-MAVS.
As indicated above, Fig 3 has been removed in the revised version of the manuscript.
- Fig 3A and related text "BTV VP3 impairs the association between MAVS and IKKε": Similarly, the relative density of flag- VP3 should be normalized by the relative density of myc-MAVS. In addition, each experiment should be at least triplicated. After statistical analysis, changes of IKKε binding ability by VP3 would be discussed. Sigle experiment is not acceptable.
Pulldown experiments have been repeated at least three times and the immunoblots presented in this article are representative of the most frequently obtained results. But, as mentioned above, Fig 3 has been removed in the revised version of the manuscript.
- In addition, according to previous studies on BTV VP3 (Kar A et al 2004 Virology....), VP3 is likely to form decamers and easily aggregate in certain condition. It would be better to confirm that VP3 was not aggregated and precipitated nonspecifically during pull-down assay.
Even if we cannot entirely rule out the possibility that nonspecific binding of VP3 occurs, we do think that we provided strong data showing specific interaction of VP3 with its dedicated partners. Indeed, Fig. 2B clearly shows that VP3 does not precipitate TRAF3 and IRF3 and only residual TBK1 binding was found.
- 4 and related text: Why the authors chose VP4 and VP7? As NS3 is likely to be IFN antagonist, NS3 should be included.
We aimed to compare VP3 with other viral proteins, including another structural protein (e.g. VP7), that are supposed to be unrelated to the IFN pathway.
- 5 and related text: The MAVS deletion mutant 1-446 did not exist in ref#13.
This mutant has been provided by Eliane Meurs (Institut Pasteur).
- Lanes 235-237: The authors logically failed to describe why they tried to confirm if VP3 could bind to K500 of MASV. MASV, "not IKKε", was K63-linked polyubiquitinated at K500 and IKKε interacted with the polyubiquitin on K500 (ref#20). Please explain why the authors thought that the ubiquitination of VP3 could be involved in MAVS interaction, logically.
As we do not have sufficiently consolidated data on this point at this stage, it has been withdrawn from the results section, and Fig 5C has been removed.
- Lane 204: Ref#19 does not show "the recruitment domain of IKKε". There may be more misusages of references. Please check carefully.
This reference has been removed.
- Discussion contains backgrounds, which should be written in Introduction, extensive repetition of the Results section, extensive future works, incorrect interpretation of results and full of speculations. Please provide much more interpretation of the results in relation to previously published work.
The text in the revised manuscript has been deeply modified.
Reviewer 4 Report
This study by Pourcelot et al. deals with the very interesting and timely matter of the functional interactions of BTV with the "infected-cell" component of the innate immune response. The authors propose a model in which binding of VP3 to MAVS impairs the recruitment of IKKepsilon; blocking in this manner the MAVS signaling pathway. While the model is interesting, further data/analysis are required for strengthening the conclusions of the study.
Specific comments:
While figure 1 convincingly shows the dose dependent interference of VP3 expression on IFN-beta induction elicited by NRIG-I, a more biologically relevant mode of activation (minimally, poly I:C; or viral infection) would considerably enhance the significance of the data. Moreover, activation via a different pathway (e.g. LPS or cGAS/STING activator) should not be affected, and can serve as a good control. In addition, if possible, showing that the inhibitory effect is occurring directly at the level of IRF3 phosphorylation would further strengthen the authors conclusions.
The statistical significance of the differences reported in 2A is not clear. In 2B, MAVS-Flag seems to bind also to GST alone (noted by authors in the text), while the amount of binding to GST-VP3 is higher a less saturated IB Flag is necessary in order to conclude that this does not stem from differences in expression. The binding of IKKepsilon seems very convincing, and could possible explain the inhibitory effects observed by the authors (not explored further in the paper and not addressed in the discussion).
The data in Fig. 3A is hard to understand: (i) the dose-dependent effect of VP3 expression is not clear, (ii) while less VP3 and IKK are precipitated in the last lane, there is also less myc-MAVS. Some form of quantification of 3B (e.g. from repeated experiments) would also be helpful in determining the significance of the observed phenomenon.
In Fig. 4 it would be interesting to see similar fluorescence analysis carried out with IKKepsilon (with VP3 and or with MAVS).
Author Response
Reviewer #3
Comments and Suggestions for Authors
This study by Pourcelot et al. deals with the very interesting and timely matter of the functional interactions of BTV with the "infected-cell" component of the innate immune response. The authors propose a model in which binding of VP3 to MAVS impairs the recruitment of IKKepsilon; blocking in this manner the MAVS signaling pathway. While the model is interesting, further data/analysis are required for strengthening the conclusions of the study.
Specific comments:
While figure 1 convincingly shows the dose dependent interference of VP3 expression on IFN-beta induction elicited by NRIG-I, a more biologically relevant mode of activation (minimally, poly I:C; or viral infection) would considerably enhance the significance of the data.
We agree with Reviewer #3 that other agonists could have been used to measure the impact of VP3 on IFN production but we focus on overexpression of the RIG-I pathway as this allowed us to previously identified the potential inhibitory effect of this viral protein (Chauveau et al., J Virol 2013).
Moreover, activation via a different pathway (e.g. LPS or cGAS/STING activator) should not be affected, and can serve as a good control. In addition, if possible, showing that the inhibitory effect is occurring directly at the level of IRF3 phosphorylation would further strengthen the authors conclusions.
Again, we agree with Reviewer #3 that measurement of additional IFN activation steps, like IRF3 phosphorylation, might strengthen the data presented here. This could be addressed in a future work.
The statistical significance of the differences reported in 2A is not clear.
The luminescence values obtained with the RRS and VP3 protein were used to define a confidence interval. We considered a protein pair as interacting if the NLR was above the previously defined threshold of 3.5 (Cassonnet et al, Nat Methods, 2011) upper limit of the confidence interval calculated from the RRS (15 in the present study).
In 2B, MAVS-Flag seems to bind also to GST alone (noted by authors in the text), while the amount of binding to GST-VP3 is higher a less saturated IB Flag is necessary in order to conclude that this does not stem from differences in expression.
For this experiment, we tried to be as precise as possible, without distorting the nature of the results. Indeed, a residual binding of MAVS to GST remained detectable, despite the different exposures used. It is important to note that the quantities of GST and GST-VP3 are comparable in the precipitated samples.
The binding of IKKepsilon seems very convincing, and could possible explain the inhibitory effects observed by the authors (not explored further in the paper and not addressed in the discussion).
The features of VP3-IKKe interaction would be addressed in more details in a future work.
The data in Fig. 3A is hard to understand: (i) the dose-dependent effect of VP3 expression is not clear, (ii) while less VP3 and IKK are precipitated in the last lane, there is also less myc-MAVS. Some form of quantification of 3B (e.g. from repeated experiments) would also be helpful in determining the significance of the observed phenomenon.
We agree with Reviewer#3 that the amounts of MAVS appear sometimes reduced in the presence of VP3. However, the total amount of MAVS expressed ectopically or at the endogenous level is not affected by VP3 overexpression (data not shown). Therefore, it seems that this finding is mostly related to the pulldown assay. We agree that this result may influence the interpretation of the effect of VP3 on the MAVS-IKKe complex. Indeed, VP3 can disrupt the formation of this complex and consequently induce the degradation of one and/or another of these actors who no longer find protection within this protein formation. We can also imagine that VP3 induces a change in conformation or a post-translational modification altering the stability of MAVS. Finally, VP3 could also recruit other factors that would play a role in protein stability. As the data presented in Fig 3 may appear not enough convincing and require further investigations, this figure has been removed from the revised version of the manuscript.
In Fig. 4 it would be interesting to see similar fluorescence analysis carried out with IKKepsilon (with VP3 and or with MAVS).
This would be addressed in a future work.
Round 2
Reviewer 1 Report
In this new revised version, Porcelot and colleagues describe the interaction between VP3 BTV and MAVS, with no further mechanistic studies. Whether this interaction is responsible for the inhibitory capacity of VP3 over IFNbeta production is still not studied.
For the current review, the authors have not performed any new experiment. Instead, the conflicting information regarding my main concerns 1 and 4 have been removed instead of addressed. When asked for further mechanistic insights (major concern 2), they refer to a future study. Only major concern 3 has been attended due to the lack of a specific tool to do it. Therefore, my comments to improve the manuscript are still unsolved.
Author Response
Reviewer 1
Open Review
(x) I would not like to sign my review report
( ) I would like to sign my review report
English language and style
( ) Extensive editing of English language and style required
( ) Moderate English changes required
(x) English language and style are fine/minor spell check required
( ) I don't feel qualified to judge about the English language and style
|
Yes |
Can be improved |
Must be improved |
Not applicable |
|
|
Does the introduction provide sufficient background and include all relevant references? |
(x) |
( ) |
( ) |
( ) |
|
Is the research design appropriate? |
(x) |
( ) |
( ) |
( ) |
|
Are the methods adequately described? |
(x) |
( ) |
( ) |
( ) |
|
Are the results clearly presented? |
(x) |
( ) |
( ) |
( ) |
|
Are the conclusions supported by the results? |
( ) |
( ) |
(x) |
( ) |
Comments and Suggestions for Authors
In this new revised version, Porcelot and colleagues describe the interaction between VP3 BTV and MAVS, with no further mechanistic studies. Whether this interaction is responsible for the inhibitory capacity of VP3 over IFNbeta production is still not studied.
For the current review, the authors have not performed any new experiment. Instead, the conflicting information regarding my main concerns 1 and 4 have been removed instead of addressed. When asked for further mechanistic insights (major concern 2), they refer to a future study. Only major concern 3 has been attended due to the lack of a specific tool to do it. Therefore, my comments to improve the manuscript are still unsolved.
The manuscript has now been to be considered as Short communication. We think that this format will be more suitable to publish our data.
Reviewer 3 Report
In the revised manuscript, the authors withdrew several data and their allegations, resulting in the loss of novelty. Unfortunately, the data shown in this manuscript is now not enough to submit as "Original Article".
Major comments
- Page 4 lanes 138-140 "In parallel, ......response pathway (Fig 1B)": ISGs production, which should be stimulated by type I interferon (IFN-I), is the downstream of IFN signaling. Thus, once IFN induction is blocked, ISGs production is also blocked. To prove "BTV VP3 inhibition of IFN-induced pathway", ISG56 promoter activation should be stimulated by IFN-I, not NRIG-I. Please consider the sentence.
- Fig2A, Pages 2-5, lanes 90-96, 161-169: The authors are likely to still misunderstand what is a random reference set (RRS). In the previous study (ref#24), regarding interaction of human papillomavirus E6 and E7 with ubiquitin proteasome system (UPR) factors, RRS (n=10) containing 9 "supposed nonbinder proteins" was first confirmed the disability of interaction with either E6 or E7 by comparison with a positive reference set (PRS) containing "7 or 5 well-established binding partners"(ref#24-Fig2). Then, the RRS (for E6 or E7) was used to determine the upper limit of the confidence interval (ref#24-Table S4). Thus, when UPR factors were tested for their binding abilities to either E6 or E7, the 9 of RRS proteins were "confirmed as nonbinder proteins". On the other hand, in this manuscript, Gluc2-fused BTV VP3 was not tested against well-established binding partners. Therefore, RRS containing 14 supposed nonbinder proteins was never confirmed its disassociation with BTV VP3. Without the confirmation, it is not suitable to use this RRS (for BTV VP3) to define the upper limit of the confidence interval. The referee understands that there is no well-established binding host-partners for BTV VP3. However, instead of host factors, BTV NS2, VP7 and VP6 might be used as PRS. If the authors are very confident that RRS used in this study did not interact with BTV VP3, please show the confidence interval determination, which contains name and NLR value of each RRS protein, mean, std (SD), and so on, as a supplemental data. In addition, there are likely several misusages of statistical terms. The positivity threshold (likely meaning the borderline between "unvalidated" and "validated" in this study) is likely the upper limit of the confidence interval in this study. Therefore, the formula for the positivity threshold should be "meant (CI:97%)", not "average ". As the authors described only in answers for the reviewers' comments, NLR threshold is 3.5 based on previous study (ref#25), which should be written in the manuscript. Further, "actual value" of the upper limit of the confidence interval in this study should be indicated. "around 15" or "above 15" are not acceptable.
- Fig3 and related text: Since GST residually bound to MAVS, the authors decided to use N-terminal flag-tagged VP3 and c-myc-tagged MAVS. In addition, to further confirm the interaction of VP3 with MAVS, they demonstrated that endogenous MAVS could interact with flag-tagged VP3. Thus, it would be better to move the text regarding Fig3 to section 3.2. In addition, they also tested the interaction of NS2 and VP7 with MAVS because NS2 and VP7 did not inhibit RIG-I-mediated IFN promoter activity in the previous study (ref#20), which should be written in the text.
- Page 6, lanes 187-189 "As the recruitment....and IKKε": As the authors deleted the data regarding attempts to identify the shared binding motif between VP3 and IKKε, the sentence should be modified.
- Page 6, lane 186 "3.3 Characterization....": As the authors did not characterized the VP3-MAVS interaction so much, the title of this section should be changed.
- Discussion still contains backgrounds, which should be written in Introduction (or just deleted), extensive repetition of the Results section, extensive future works and full of speculations.
- Page 9 lanes 246-248 "In BTV......with MAVS.": In this study, only tagged VP3, which is singly expressed with host factors, was used. Thus, there is a big gap between "VP3-NS2 interaction in BTV-infection" and "tagged VP3-MAVS interaction in transient expressing system". Please consider the sentences.
- Page 9 lanes 261-263 "In particular.....ubiquitination (data not shown)": So far, "the involvement of ubiquitination of TRAFs-related proteins in RIG-I-mediated IFN induction" and "the ubiquitination of VP3" is completely different story. Please logically describe why they think that the role of ubiquitination in protein binding should be addressed with clearly indicating the connection between "ubiquitination of VP3" and "RIG-I-mediated IFN induction".
- There are still logical fallacies as well as minor errors such as typos etc. Please go through the whole manuscript carefully.
Author Response
Reviewer 3
Open Review
(x) I would not like to sign my review report
( ) I would like to sign my review report
English language and style
( ) Extensive editing of English language and style required
(x) Moderate English changes required
( ) English language and style are fine/minor spell check required
( ) I don't feel qualified to judge about the English language and style
|
Yes |
Can be improved |
Must be improved |
Not applicable |
|
|
Does the introduction provide sufficient background and include all relevant references? |
(x) |
( ) |
( ) |
( ) |
|
Is the research design appropriate? |
( ) |
( ) |
(x) |
( ) |
|
Are the methods adequately described? |
( ) |
(x) |
( ) |
( ) |
|
Are the results clearly presented? |
( ) |
(x) |
( ) |
( ) |
|
Are the conclusions supported by the results? |
( ) |
( ) |
(x) |
( ) |
Comments and Suggestions for Authors
In the revised manuscript, the authors withdrew several data and their allegations, resulting in the loss of novelty. Unfortunately, the data shown in this manuscript is now not enough to submit as "Original Article".
Major comments
- Page 4 lanes 138-140 "In parallel, ......response pathway (Fig 1B)": ISGs production, which should be stimulated by type I interferon (IFN-I), is the downstream of IFN signaling. Thus, once IFN induction is blocked, ISGs production is also blocked. To prove "BTV VP3 inhibition of IFN-induced pathway", ISG56 promoter activation should be stimulated by IFN-I, not NRIG-I. Please consider the sentence.
We agree with Reviewer#3. We did not specifically address the action of VP3 on the IFN signaling (Jak/STAT) pathway upon stimulation by IFN-I. Here, the ISG56 promoter was used as a second readout of the consequence of VP3 on IFN induction. A sentence has been modified: “VP3 is able to impair activation of specific promoters along the entire IFN pathway induced upon activation of the RLR signalling cascade.” (lanes 152-153).
- Fig2A, Pages 2-5, lanes 90-96, 161-169: The authors are likely to still misunderstand what is a random reference set (RRS). In the previous study (ref#24), regarding interaction of human papillomavirus E6 and E7 with ubiquitin proteasome system (UPR) factors, RRS (n=10) containing 9 "supposed nonbinder proteins" was first confirmed the disability of interaction with either E6 or E7 by comparison with a positive reference set (PRS) containing "7 or 5 well-established binding partners"(ref#24-Fig2). Then, the RRS (for E6 or E7) was used to determine the upper limit of the confidence interval (ref#24-Table S4). Thus, when UPR factors were tested for their binding abilities to either E6 or E7, the 9 of RRS proteins were "confirmed as nonbinder proteins". On the other hand, in this manuscript, Gluc2-fused BTV VP3 was not tested against well-established binding partners. Therefore, RRS containing 14 supposed nonbinder proteins was never confirmed its disassociation with BTV VP3. Without the confirmation, it is not suitable to use this RRS (for BTV VP3) to define the upper limit of the confidence interval. The referee understands that there is no well-established binding host-partners for BTV VP3. However, instead of host factors, BTV NS2, VP7 and VP6 might be used as PRS. If the authors are very confident that RRS used in this study did not interact with BTV VP3, please show the confidence interval determination, which contains name and NLR value of each RRS protein, mean, std (SD), and so on, as a supplemental data. In addition, there are likely several misusages of statistical terms. The positivity threshold (likely meaning the borderline between "unvalidated" and "validated" in this study) is likely the upper limit of the confidence interval in this study. Therefore, the formula for the positivity threshold should be "meant (CI:97%)", not "average ". As the authors described only in answers for the reviewers' comments, NLR threshold is 3.5 based on previous study (ref#25), which should be written in the manuscript. Further, "actual value" of the upper limit of the confidence interval in this study should be indicated. "around 15" or "above 15" are not acceptable.
We understand the Reviewer#3’s point of view. As the manuscript appeared now as a Short communication, supplemental data are not really suitable for this format. However, raw data can be provided upon further request. The list of the 14 RRS is as follows: APOOL, CACG7, CNTN2, DBH, DPYSL2, GSTT1, GYPAL, LRRC2, FFE2L1, NXPH1, PLEKHA9, SLC7A13, UGT3A1, TRPT1. According to our collaborators, the previously selected RRS data set has been designed to be applicable to further studies. The exact calculated NLR threshold (15.2) has been added in the text as well as the exact positive values for MAVS and IKKe (17.4 and 17.7 respectively). The formula has been changed accordingly.
- Fig3 and related text: Since GST residually bound to MAVS, the authors decided to use N-terminal flag-tagged VP3 and c-myc-tagged MAVS. In addition, to further confirm the interaction of VP3 with MAVS, they demonstrated that endogenous MAVS could interact with flag-tagged VP3. Thus, it would be better to move the text regarding Fig3 to section 3.2. In addition, they also tested the interaction of NS2 and VP7 with MAVS because NS2 and VP7 did not inhibit RIG-I-mediated IFN promoter activity in the previous study (ref#20), which should be written in the text.
We are grateful to Reviewer#3 for these pertinent comments. The manuscript has now been extensively amended to adapt it to a Short communication format. The text regarding Fig3 is no more detached in a distinct section (lanes 203-208).
We also took into account the remark of Reviewer#3 regarding the viral controls and we added this sentence in the revised version (lanes 201-203): “VP7 and NS2 were used as controls, as they did not inhibit RIG-I-mediated IFN promoter activity and are therefore not supposed to interact with components of the IFN pathway”.
- Page 6, lanes 187-189 "As the recruitment....and IKKε": As the authors deleted the data regarding attempts to identify the shared binding motif between VP3 and IKKε, the sentence should be modified.
The sentence has been removed in the new version.
- Page 6, lane 186 "3.3 Characterization....": As the authors did not characterized the VP3-MAVS interaction so much, the title of this section should be changed.
As mentioned above, the text has been changed to fit with Short communication format. Thus, in this revised version, section titles have been removed.
- Discussion still contains backgrounds, which should be written in Introduction (or just deleted), extensive repetition of the Results section, extensive future works and full of speculations.
As suggested by Reviewer#3, the text has been deeply modified. Backgrounds from the Discussion section have been moved to the Introduction and the text reduced to be adapted to Short communication.
- Page 9 lanes 246-248 "In BTV......with MAVS.": In this study, only tagged VP3, which is singly expressed with host factors, was used. Thus, there is a big gap between "VP3-NS2 interaction in BTV-infection" and "tagged VP3-MAVS interaction in transient expressing system". Please consider the sentences.
We agree with Reviewer#3 that this sentence is a bit confusing and can be misinterpreted. It has been moved to the Introduction where VP3 functions are described (lanes 63-66 of the new version).
- Page 9 lanes 261-263 "In particular.....ubiquitination (data not shown)": So far, "the involvement of ubiquitination of TRAFs-related proteins in RIG-I-mediated IFN induction" and "the ubiquitination of VP3" is completely different story. Please logically describe why they think that the role of ubiquitination in protein binding should be addressed with clearly indicating the connection between "ubiquitination of VP3" and "RIG-I-mediated IFN induction".
We agree with Reviewer#3 that the role of ubiquitination in the VP3 story is too speculative at this stage and this part has been removed in the new version of the manuscript.
- There are still logical fallacies as well as minor errors such as typos etc. Please go through the whole manuscript carefully.
As suggested by Reviewer#3, we checked again the whole manuscript carefully and we apologize if some minor errors still remain.
Reviewer 4 Report
The revised version of the manuscript is nearly identical to the previous one. The authors chose not to perform any of the suggested control experiments, which could reinforce the data interpretation; stating essentially that they will be subject of future study.
The text additions (surely stemming from comments of other reviewers) make certain aspects of this study clearer.
Author Response
Reviewer 4
Open Review
(x) I would not like to sign my review report
( ) I would like to sign my review report
English language and style
( ) Extensive editing of English language and style required
( ) Moderate English changes required
(x) English language and style are fine/minor spell check required
( ) I don't feel qualified to judge about the English language and style
|
Yes |
Can be improved |
Must be improved |
Not applicable |
|
|
Does the introduction provide sufficient background and include all relevant references? |
(x) |
( ) |
( ) |
( ) |
|
Is the research design appropriate? |
( ) |
(x) |
( ) |
( ) |
|
Are the methods adequately described? |
(x) |
( ) |
( ) |
( ) |
|
Are the results clearly presented? |
( ) |
(x) |
( ) |
( ) |
|
Are the conclusions supported by the results? |
( ) |
(x) |
( ) |
( ) |
Comments and Suggestions for Authors
The revised version of the manuscript is nearly identical to the previous one. The authors chose not to perform any of the suggested control experiments, which could reinforce the data interpretation; stating essentially that they will be subject of future study.
The text additions (surely stemming from comments of other reviewers) make certain aspects of this study clearer.
The manuscript has now been to be considered as Short communication. We think that this format will be more suitable to publish our data.